# The Use of Rapid COVID-19 Antigen Test in the Emergency Department as a Decision-Support Tool

**DOI:** 10.3390/microorganisms11020284

**Published:** 2023-01-21

**Authors:** Lilac Meltzer, Sharon Amit, Mayan Gilboa, Ilana Tal, Bella Mechnik, Avi Irony, Hindi Engelrad, Avi Epstein, Yael Frenkel-Nir, Yuval Levy, Yitshak Kreiss, Gili Regev-Yochay

**Affiliations:** 1Infection Control & Prevention Unit, Sheba Medical Center, Ramat Gan 52621, Israel; 2Sackler School of Medicine, Tel Aviv University, Tel Aviv 69978, Israel; 3Clinical Microbiology, Sheba Medical Center, Ramat Gan 52621, Israel; 4Emergency Department, Sheba Medical Center, Ramat Gan 52621, Israel; 5The General Management, Sheba Medical Center, Ramat Gan 52621, Israel

**Keywords:** COVID-19, SARS-CoV-19, antigen, rapid tests, emergency department, triage, triage protocol, decision-making

## Abstract

The emergency department (ED) is the initial point of contact between hospital staff and patients potentially infected with SARS-CoV-2, thus, prevention of inadvertent exposure to other patients is a top priority. We aimed to assess whether the introduction of antigen-detecting rapid diagnostic tests (Ag-RDTs) to the ED affected the likelihood of unwanted SARS-CoV-2 exposures. In this retrospective single-center study, we compared the rate of unwarranted exposure of uninfected adult ED patients to SARS-CoV-2 during two separate research periods; one before Ag-RDTs were introduced, and one with Ag-RDT used as a decision-support tool. The introduction of Ag-RDTs to the ED significantly decreased the relative risk of SARS-CoV-2-negative patients being incorrectly assigned to the COVID-19 designated site (“red ED”), by 97%. There was no increase in the risk of SARS-CoV-2-positive patients incorrectly assigned to the COVID-19-free site (“green ED”). In addition, duration of ED admission was reduced in both the red and the green ED. Therefore, implementing the Ag-RDT-based triage protocol proved beneficial in preventing potential COVID-19 nosocomial transmission.

## 1. Introduction

The COVID-19 pandemic in Israel has resulted thus far (June 2022) in 4.3 million infections and over 10,000 deaths, out of a population of 9.2 million people [1]. Early during the pandemic, much effort was invested in separating COVID-19-infected patients from uninfected patients, to protect the uninfected and prevent nosocomial outbreaks. SARS-CoV-2-positive patients were treated in COVID-19 designated sites, where healthcare workers (HCWs) used heavy personal protective equipment (PPE), to minimize the risk of infection. At these early stages, the inability to differentiate between infected and uninfected patients upon their hospital arrival affected the quality of care, morbidity, and mortality [2]. The massive use of PPE, which was necessary to protect HCWs from nosocomial infections, resulted in sub-optimal clinical performance and impaired communication between HCWs and patients [3].

As the ED is the initial point of contact of hospital staff with patients potentially infected with SARS-CoV-2, fast and accurate diagnostic tools for COVID-19 at the ED are essential [4]. Real-time Reverse Transcription-Polymerase Chain Reaction (RT-PCR) is considered the gold standard for detecting COVID-19 but requires several hours from sampling to result, especially during disease surges when personnel and lab resources are limited [5]. Conversely, point-of-care antigen-detecting rapid diagnostic tests (Ag-RDTs), have a turnaround time of 15 min, require minimal training and infrastructure, and are cheap and easy to use. Nevertheless, Ag-RDTs are not as sensitive as RT-PCR tests and cannot replace them as a definite diagnostic tool for COVID-19 [6]. Here, we report how the introduction of Ag-RDTs as a decision support tool in the ED COVID-19 triage protocol reduced patient risk for inadvertent COVID-19 exposure and potential nosocomial transmission.

## 2. Materials and Methods

### 2.1. Ethical Considerations

The research protocol was approved by the local Institutional Review Board. Verbal informed consent was obtained from the study participants.

### 2.2. Setting

The Sheba Medical Center (SMC) is the largest tertiary hospital in Israel, with over 1900 acute and long-term hospital beds. The main ED at SMC had approximately 140,000 admissions annually in the pre-COVID-19 era.

### 2.3. Study Period

In Israel, the COVID-19 pandemic has resulted to date in five distinct COVID-19 surges. The study compared data from two distinct periods: period 1, during the second COVID-19 surge (21 August 2020–4 October 2020) and before the introduction of Ag-RDTs to the ER, and period 2, during the third COVID-19 surge (7 December 2020–21 March 2021) during which Ag-RDTs were fully implemented as a decision support tool (Figure 1).

### 2.4. Study Population

The study population consisted of all adults (age ≥ 18 years) ED patients within the study period, who had a valid PCR result from their ED visit. Patient information collected from the hospital electronic database included socio-demographic details, as well as ED department allocation, hospitalization status, RT-PCR test results, and Ag-RDT test results if available. 

### 2.5. Triage Strategy

Early in the COVID-19 pandemic, the SMC opened a separate biological (“red”) ED for all suspected or detected COVID-19 cases. Thus, two separate ED sites operated simultaneously: a regular non-COVID-19 ED (“green ED”) and a suspected COVID-19 ED (“red ED”) as described previously [7]. Before 5th October 2020, during period 1, the classification method of patients suspected of COVID-19 was based on a symptomatic case definition, as well as a history of exposure to a positive patient, arrival from abroad or from hyperendemic sites within Israel, and patients who were confirmed as COVID-19-positive prior to their arrival at SMC. Clinical suspicion was confirmed according to PCR results which were available within 6–8 h (Figure 2a). On 5th October 2020, Ag-RDTs were introduced to the ED as a decision-support tool for the initial triage of all COVID-19 suspected cases. Thus, in period 2, all patients who were classified as being suspected of COVID-19 by the criteria mentioned above underwent an Ag-RDT. If the result was negative, they were admitted to the green ED, whereas a positive result led to admission to the red ED (Figure 2b).

### 2.6. Sample Collection and Analysis

A nasopharyngeal swab sample was obtained by trained personnel following appropriate safety precautions. The Ag-RDT kit used as the decision-support tool was the Nowcheck COVID-19 Ag test (Bionote, Seoul, South Korea), according to the manufacturer’s specifications. In addition, nasal and oropharyngeal samples were obtained and tested for SARS-CoV-2 by an RT-PCR test, as previously described [8]. Positive results were further stratified according to the cycle threshold (Ct) value, serving as a surrogate for viral load. Patients whose Ct value was below 30 were defined as presumably infectious [9,10].

### 2.7. Statistical Analysis

The main outcomes assessed were the sensitivity and specificity of the triage policy. Hence, the risk of uninfected SARS-CoV-2 ED patients being exposed to SARS-CoV-2, due to misassignment to the red ED, and the risk of inadvertent exposures due to misassigning SARS-CoV-2 infected patients to the green ED. Sensitivity, specificity, PPV, and NPV were calculated for the triage process in each period, with or without the use of Ag-RDT, compared with RT-PCR results. Average daily patient numbers for the red and green ED were calculated to compare the study periods, which differed in length and positive patient burden. The risk reduction for uninfected ED patients exposed to infected COVID-19 patients was calculated as follows: the risk in each period was defined as the proportion of eventually COVID-19-negative patients, assigned to the red ED, out of all ED patients. The risk reduction was calculated as the risk ratio between the two study periods.

Average daily patient numbers for the red and green ED were calculated to compare the study periods, which differ in length and positive patient burden. Additionally, we calculated the additional potential risk of exposing uninfected green ED patients to undetected positive COVID-19 patients due to the change in triage protocol. The risk was defined as the proportion of undetected COVID-19 patients misassigned in the green ED out of all ED patients. The risk ratio between the two periods was similarly calculated. The predicted association, as well as its 95% confidence interval (CI), were calculated using the exact binomial method; *p*-values < 0.05 were considered statistically significant.

Another outcome assessed was the impact of the new triage algorithm on the duration of ED admission. Average ED admission was calculated for each group of patients during the two study periods, and a 95% confidence interval (CI) was calculated as described above. The number of exposure hours saved due to the new triage protocol, or the extra time that patients during study period 2 would have spent in the ED, was also calculated. This was executed by multiplying the difference between average ED admission durations between periods 1 and 2 by the number of patients admitted to the ED during period 2. 

## 3. Results

### 3.1. ED Visitations

During the two study periods, a total of 18,307 patients visited the ED and were assigned to either the green or the red ED. Of those patients, 4659 visited the ED during 6 weeks of period 1 (21 August 2020–4 October 2020) and 13,648 during the 15 weeks of period 2 (7 December 2020–21 March 2021). Overall, 18,103 nasopharyngeal PCR samples were obtained from ED patients: 4511 from patients during period 1 and 13,592 from patients during period 2. Further demographic data are shown in Table 1. The disease prevalence during period 1 was 7.8% (95% CI, 7–8.6) and during period 2 was 6.5% (95% CI, 6.1–6.9). The proportion of patients with a Ct value < 30 was 5.5% (95% CI, 4.9–6.3) during period 1 and 3.6% (95% CI, 3.3–4) during period 2.

### 3.2. Risk Reduction—Red ED

We observed a significant reduction in the risk of COVID-19-negative patients being exposed to COVID-19-positive patients, and thus, a reduction in their risk of infection. The number of suspected COVID-19 patients that were eventually COVID-19-negative but assigned to the red ED, declined from 15.9 patients/day during period 1 to 0.5 patients/day during period 2 (*p* < 0.0001) (Table 2). Consequently, the risk of incorrectly assigning uninfected patients to the red ED, defined as the proportion of COVID-19-negative patients assigned to the red ED from all ED patients, decreased from 15.8% (95% CI: 14.8–16.9%) during period 1 to 0.4% (95% CI: 0.3–0.5%) during period 2, with a relative risk reduction of 97%. During period 1, 714 COVID-19-negative patients were assigned to the red ED.

Despite the significant decline in the misassignment of COVID-19-negative patients to the red ED during period 2, 50 patients were still incorrectly assigned. Therefore, the reasons for those misassignments were examined per patient. Of those 50 misassigned patients, 22 (44%) had URI symptoms, dysgeusia or anosmia, and/or recent exposure to a COVID-19-confirmed individual, 19 (34.5%) had a recent positive PCR result taken elsewhere before ED referral and were considered recent recoverees, and in 6 (12%) nursing home residents no history could be obtained. In two additional patients (4%) who were self-referred to the ED, COVID-19 testing was performed solely at the patient’s request, and one case (2%) had a false-positive Ag-RDT result and therefore was assigned to the red ED. 

### 3.3. Risk Reduction—Green ED

The second main important outcome of this study is that the absolute number of inadvertent exposures to COVID-19-positive patients, that were misassigned to the green ED, has not changed considerably due to the new triage protocol. The average daily number of COVID-19-positive patients assigned to green ED was 1 patient during period 1 (total of 45 patients) and 1.7 patients during period 2 (total of 182 patients) (*p* = 0.567). Consequently, the proportion of COVID-19-positive patients assigned to the green ED increased from 1% (95% CI: 0.7–1.3%) during period 1 to 1.3% (95% CI: 1.1–1.5%) during period 2. Therefore, the relative risk of incorrectly assigning to the green ED was 1.5 (95% CI: 1.1 to 2) during period 2.

To further assess the risk of having a presumably infectious COVID-19 patient in the green ED, we used a CT value of less than 30 as a correlate of infectivity [9,10]. The proportion of presumably infectious patients (with Ct value < 30), decreased from 5.5% (95% CI: 4.9–6.2%) during period 1 to 3.6% (95% CI: 3.3–3.9%) during period 2. The average daily numbers of presumably infectious patients assigned to the green ED were 0.6 and 0.7 during periods 1 and 2, respectively (*p* = 0.2355). Consequently, the relative risk of incorrectly assigning COVID-19-infected and presumably infectious patients to the green ED did not change significantly (1.4, 95% CI 0.9–2.1, *p* = 0.138). 

Since one of the concerns in the use of Ag-RDT was their lower sensitivity, as a part of the risk stratification, patients with positive PCR results and CT values higher than 30 (lower risk of being infectious) were also analyzed separately. Their proportion had an insignificant increase from 2.3% (95% CI: 1.8–2.7%) to 2.9% (95% CI: 2.6–3.2%) between periods 1 and 2. The average daily numbers of these patients assigned to the green ED were 0.4 and 1.1 during periods 1 and 2, respectively (*p* = 0.0435). Consequently, the relative risk of incorrectly assigning COVID-19-infected patients Ct values > 30 (which have been shown to be less or non-infective) to the green ED was 1.4 (95% CI 0.9–2.2, *p* = 0.0897). 

### 3.4. Sensitivity, Specificity, PPV, NPV

The sensitivity of the ED triage protocol during both research periods was 87.2% (95% CI, 83.3–90.5) for period 1 and 79.3% (95% CI, 76.5–82) for period 2; while the specificity was 82.8% (95% CI, 81.7–84) and 99.6% (95% CI, 99.5–99.7), respectively. In patients with low Ct values, ED triage-protocol sensitivity decline was less pronounced and decreased from 90% (95% CI, 85.6–93.4) during period 1 to 85.6% (95% CI, 82.1–88.6). The negative predictive value (NPV) of the ED triage protocol was similar during both period 1 (98.7%, 95% CI, 98.3–99) and period 2 (98.6%, 95% CI, 98.4–98.8). However, the positive predictive value (PPV) increased from 30.1% (95% CI, 28.5–31.7) during period 1 to 93.3% (95% CI, 91.4–94.9) during period 2.

### 3.5. Duration of ED Admission

During period 1, the 4511 patients admitted to the ED spent an average of 7 h (95% CI 6.9–7.1 h) per patient at the ED. The 13,592 patients admitted to the ED during period 2 spent an average of 6.7 h (95% CI 6.6–6.8 h) per patient. When calculating “exposure hours” as described in the statistical analysis section, this amounts to 4621.3 h saved. In particular, the average ED admission among patients who were assigned to the red ED was reduced from 6.2 h (95% CI 5.9–6.5 h) during period 1 to 6 h (95% CI 5.8–6.2 h) during period 2. The average ED admission for patients assigned to the green ED was also reduced—from 7.3 h (95% CI 7.2–7.4 h) during period 1 to 6.8 h (95% CI 6.7–6.9 h) during period 2 (see Table 3).

To further assess the risk of inadvertent exposures in the green ED, patients who were misassigned to the green ED (COVID-19-positive) were divided into hospitalized and discharged. Hospitalized patients’ admission duration insignificantly increased between study periods by 3.8%—from 7.9 h to 8.2 h (95% CI 6.6–9.2 and 7.3–9 h, respectively) while discharged patients’ admission duration decreased by 15.2%—from 5.3 h to 4.6 h (95% CI 3.5–7.1 and 3.9–5.3 h, respectively).

## 4. Discussion

The COVID-19 pandemic has led to an unprecedented crisis in healthcare systems. Hospitals became the potential foci of viral exposure and transmission. Consequently, prevention of inadvertent COVID-19 exposure and the preservation of patients’ as well as hospital staff’s safety became a top priority [11,12]. At the same time, it was necessary to maintain the quality of medical care, which was affected by the inability to classify patients as infected or uninfected effectively and quickly [2]. Our study demonstrated that the addition of the COVID-19 Ag-RDT as a preliminary evaluation tool for all ED patients suspected of COVID-19 infection, resulted in a dramatic decline in the misassignment of COVID-19-negative patients to COVID-19 treatment areas, without an increased risk of incorrectly assigning infective (Ct value < 30) COVID-19-positive patients to the non-COVID-19 treatment areas [9,10,13]. For every COVID-19-positive patient who was incorrectly assigned to the green ED during period 2, 20.82 COVID-19-negative patients avoided being incorrectly assigned to the red ED, according to symptoms and/or history, as was practiced during period 1.

Previous studies of COVID-19 Ag-RDTs emphasized the tests’ diagnostic performance. Several studies showed that Ag-RDT has moderate sensitivity and high specificity [13,14], in concordance with the real-world performance of these kits demonstrated at the Sheba Medical Center [8]. Other kits differ in sensitivity, but most demonstrate high specificity [15]. There are limited data in the published literature describing the effects of adding Ag-RDTs to the triage regimen in the ED. In one retrospective cohort study, rapid tests were performed according to the clinicians’ judgment and targeted only specific patient populations (likely to be hospitalized, or unable to self-isolate). This study found that introducing COVID-19 Ag-RDTs to the ED resulted in a 65.6% reduction in the median red ED exposure time [16]. This study also demonstrates the reduction in exposure hours in both the red and the green ED, but the more significant change was indeed in the misassigned patients at the red ED—where admission length (and therefore, the exposure time) was shortened by 30.2%. However, this change may be attributed to more experience in treating COVID-19-positive patients during study period 2, and fewer patients assigned to the red ED (as demonstrated in the results).

Although other studies have also shown the benefits of introducing Ag-RDT as part of a triage protocol, our study has the largest cohort. Furthermore, even though many healthcare centers divided the COVID-19-unsuspected and COVID-19-suspected patients to different ED sites, our study best demonstrates the risk reduction for COVID-19-negative patients to be exposed to COVID-19-positive patients in either of the ED sites. In addition, the comparison between two study periods, one of them before the introduction of Ag-RDTs to the ED as part of the triage protocol, and one after, is unique to our research [17,18,19,20].

Another interesting point, which was beyond the scope of this study, was the potential economic gain following the introduction of Ag-RDTs as a decision-making tool at the ED. One study which evaluated the use of Ag-RDT at a German hospital claimed a significant cost reduction. This was mostly due to Ag-RDT’s high specificity, which resulted in a lower proportion of unnecessary bed blocking [21]. The implementation of Ag-RDT, due to its point-of-care, time-saving, and low-priced qualities, may assist in preserving crucial resources– such as hospital beds, PPE, staff hours and training, and more. Our study has several limitations. Regarding the triage method, patients who were asymptomatically infected and did not meet the criteria for PCR and/or Ag-RDT screening (as described above), and who were not hospitalized—may have never been tested. These patients may increase the number of inadvertent exposures in the green ED. Nevertheless, since the triage of these patients did not change between the two study periods—this does not change the results of the study. Furthermore, the new triage protocol might narrow the time window for inadvertent exposures to asymptomatic patients in the green ED who were not tested. In any case, untested patients in the green ED (who were not COVID-19-suspected or hospitalized)—represent the necessary compromise between the ideal vast screening and the costs and resources required for it.

Another limitation is that this is a single-center study reflecting the epidemiological conditions in Israel during a specific period. However, we believe our results are relevant for similar large tertiary centers. The clinical–epidemiological case definition utilized during period 1 was relatively broad, preferentially opting for better sensitivity than specificity and therefore resulted in a high false-positive result. This definition was similar to that recommended by the US-CDC [22] as well as other advisory bodies [23], and is relevant even today.

Furthermore, our study was conducted during the period when the Alpha variant was the predominant variant before the emergence of the Omicron variant. It has been suggested that the Ag-RDTs are less sensitive to Omicron, but this has been mostly proven wrong. Studies found that Ag-RDTs perform well as a point-of-care detection test for the Omicron variant, with sensitivity comparable to previous variants (especially for high viral load), and excellent specificity [24]. We argue that, even if Ag-RDTs will display reduced detection ability upon the emergence of new variants of concern, they may still serve as a safety net and allow additional protection for HCWs and patients in the emergency department.

## 5. Conclusions

The introduction of Ag-RDT use to the ED COVID-19 triage protocol has resulted in a dramatic decline in the risk of misassignment of suspected COVID-19 patients who were eventually negative, into COVID-19 designated sites. This was managed without a reciprocal increase in the risk of positive infective COVID-19 patients being assigned to the COVID-19-free, green ED. Thus, this triage protocol potentially decreased patients’ risk of COVID-19 nosocomial infection, while improving the medical care that patients receive. In addition, as contact precautions and PPE are known to affect the quality of care, early identification of COVID-19-positive patients allows HCWs to avoid unnecessary PPE and potentially improves the quality of care at the ED [25]. Considering the shortening of triage time and overcrowding of the ED, the introduction of Ag-RDTs as part of the triage, allowed for better streamlining of tailored medical services for patients in the COVID-19 designated site and the COVID-19-free site alike.

## Figures and Tables

**Figure 1 microorganisms-11-00284-f001:**
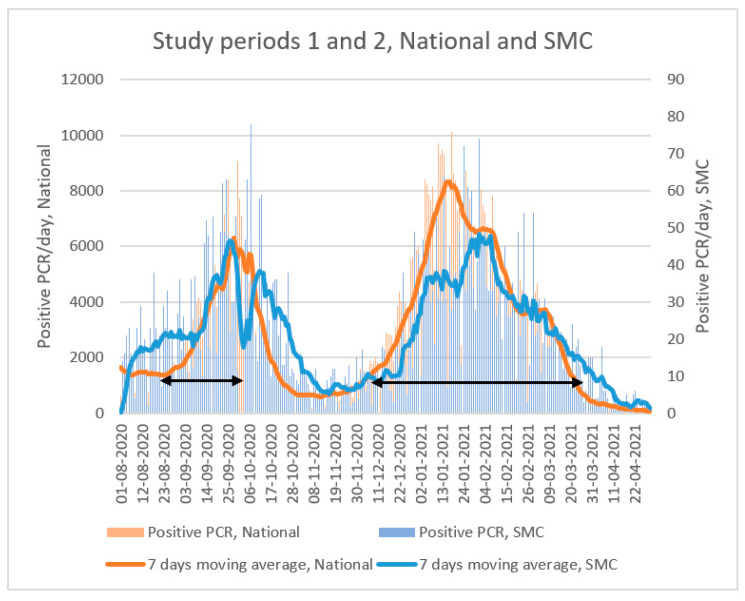
Daily detected SARS-CoV-2 cases in Israel and Sheba Medical Center during the study periods. The black arrows indicate study periods 1 (21 August 2020–4 October 2020) and 2 (7 December 2020–21 March 2021).

**Figure 2 microorganisms-11-00284-f002:**
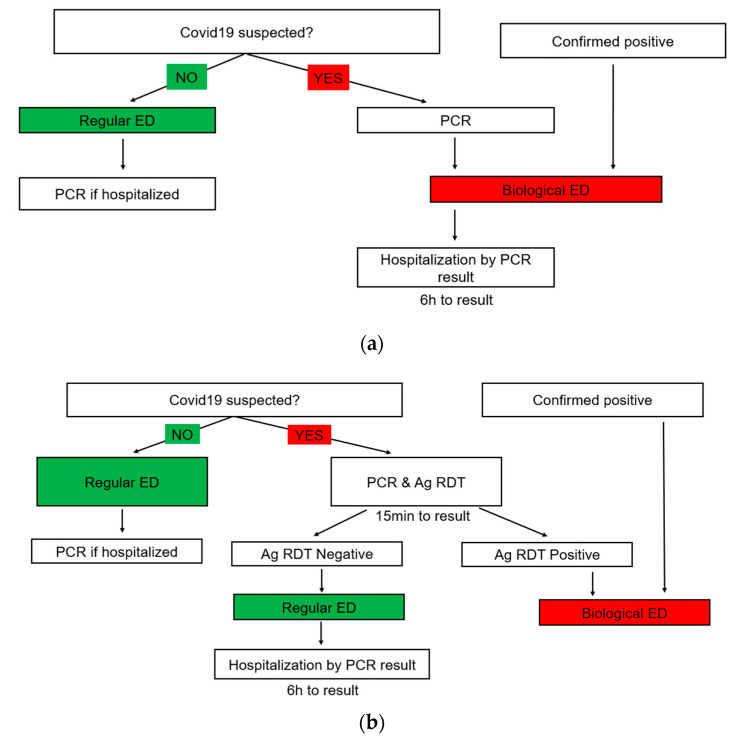
(**a**) The triage protocol during period 1, prior to the introduction of Ag-RDTs to the ED. (**b**) The triage protocol during period 2, while Ag-RDT was used as a decision-support tool.

**Table 1 microorganisms-11-00284-t001:** Study population.

	Period 1 (21 August 2020–4 October 2020)		Period 2 (7 December 2020–21 March 2021)	
	N	%	N	%
ED patients	4659		13,648	
ED patients with PCR test result	4511		13,592	
Average ED patient with PCR test result/day	100.24		129.45	
Sex (Male)	2442	54.1%	7021	51.7%
Age—Mean	61		61	
Age—Median (range)	65 (18–104)		66 (18–105)	
Hospitalized	3717	82.4%	10,342	76.1%
Number of Ag tests	0		7922	

**Table 2 microorganisms-11-00284-t002:** The daily average and percentage of ED patients assigned to the green/red EDs during research periods 1 and 2.

	Period 1 (21 August 2020–4 October 2020)		Period 2 (7 December 2020–21 March 2021)	
	Green	Red	Green	Red
COVID-19 Positive ED patients	1 (1%)	6.8 (6.8%)	1.7 (1.3%)	6.7 (5.1%)
COVID-19 Positive Patients, CT < 30	0.6 (0.6%)	5 (5%)	0.7 (0.5%)	3.9 (3%)
COVID-19 Negative Patients	75.6 (76.4%)	15.9 (15.8%)	120.6 (93.2%)	0.5 (0.4%)
Total ED Patients	77.6 (77.4%)	22.7 (22.6%)	123.9 (94.5%)	7.1 (5.5%)

**Table 3 microorganisms-11-00284-t003:** Mean ED admission length (hours), change percentage between two study periods, and exposure time saved the daily average and percentage of ED patients assigned to the green/red EDs during research periods 1 and 2.

	Period 1 Mean ED Admission, Hours (95% CI)	Period 2 Mean ED Admission, Hours (95% CI)	% Change	Exposure Time Saved, Hours
All Patients	7 (6.9–7.1)	6.7 (6.6–6.8)	4.8	4621.3
Green ED	7.3	6.8	7.7	7192.1
Red ED	6.2	6	3.4	157.3
Misassigned to green ED	6.9 (5.8–8)	6.7 (6.1–7.3)	2.9	36.4
Misassigned to green ED, CT < 30	5.9 (4.8–7)	5.5 (4.8–6.1)	6.5	26.6
Misassigned to red ED	6 (5.7–6.3)	4.2 (3.6–4.8)	30.2	90.5

## Data Availability

Data supporting reported results can be provided upon request.

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
