# Peer review of "The Use of Rapid COVID-19 Antigen Test in the Emergency Department as a Decision-Support Tool"

_microorganisms, 2023, doi:10.3390/microorganisms11020284_

Round 1
Reviewer 1 Report
This paper explores the use of antigenic rapid diagnostic tests (Ag-RDT) for triage of patients between the green (intended Covid negative) and red (intended Covid positive) emergency wards. The paper focuses on the number of patients mis-assigned to green or red, as determined by their PCR results. After introduction of Ag-RDT, the rate at which negative patients were wrongly assigned to the red ward dropped substantially. The rate at which positive patients were wrongly assigned to the green ward did not change significantly.
Comment:
An additional limitation of the study not mentioned here is that some individuals who are admitted to the green ED might be asymptomatically infected: if they don't show clinical signs, don't have a compelling history of exposure, and aren't hospitalised, they may have never been tested. This shouldn't change the results of the study, unless the triage of these individuals was different between the two time periods.
Minor comments/typos:
Line 96
What does the "(1)" refer to? There is no "(2)"
Line 134
Sentence is a bit scrambled and so ambiguous
Figure 2b
I don't really understand what the "Confirmed positive" box flowing into the Red ED refers to. It looks like some patients are coming directly into the red ED having been confirmed positive elsewhere, but this isn't mentioned in the Triage Strategy section, line 82.
Also, it's a bit unclear what hospitalisation means within the triage strategy: from the diagram, if patients are admitted to the green ward, they are not tested, unless they are hospitalised. What is going on with patients who are admitted but not hospitalised? Are they there long enough to transmit to other hospital users? Please describe in more detail.
Line 229
Spelling: "Mis-assignment"
Author Response
We thank you for allowing us to revise the article, and we have carried out revisions following the reviewers’ comments. Please see our responses below, addressing each comment raised by each reviewer:
Reviewer 1 comments:
An additional limitation of the study not mentioned here is that some individuals who are admitted to the green ED might be asymptomatically infected: if they don't show clinical signs, don't have a compelling history of exposure, and aren't hospitalized, they may have never been tested. This shouldn't change the results of the study unless the triage of these individuals was different between the two time periods.
This is correct. Patients who were asymptomatically infected and did not meet the criteria for PCR and/or Ag-RDT screening (as described above), and who weren’t hospitalized – may have never been tested. But this was similar in the two periods. So it should not have changed the results of the study. In any case, to clarify this point, we added this information to the discussion section (now lines 274-283).
Minor comments/typos:
Line 96
What does the "(1)" refer to? There is no "(2)"
This notation was indeed out of place, it was deleted, and the sentence was rephrased (now line 99).
Line 134
Sentence is a bit scramThe sentence so ambiguous
We agree, the relevant sentence was rephrased to clarify (now line 146).
Figure 2b
I don't really understand w "Confirmed positive" box flowing into the Red ED refers to. It looks like some patients are coming directly into the red ED having been confirmed positive elsewhere, but this isn't mentioned in the Triage Strategy section, line 82.
In both study periods, individuals who were confirmed positive by PCR prior to their ED admission were referred to the red ED. Since this part of the triage method hasn’t changed, between the two study periods, to our understanding, this doesn’t affect our results. We changed the figures to demonstrate that and added a clarification to section 2.5 – triage strategy (now line 79).
Also, it's a bit unclear what hospitalization means within the triage strategy: from the diagram, if patients are admitted to the green ward, they are not tested, unless they are hospitalized. What is going on with patients who are admitted but not hospitalized? Are they there long enough to transmit to other hospital users? Please describe in more detail.
Patients who were not COVID-19-suspected and were not hospitalized following their ED admission – were not tested. These patients were not part of this study’s cohort, and there are no data regarding their COVID-19 status post their ED admission. Although they did stay long enough at the green ED to cause inadvertent exposures – this time was shorter in period 2 compering to period 1. We added the relevant data to the results section and discussed it (lines 247-283). We also acknowledged that it would have been best to test these patients also, but it would require further resources, that were unavailable.
Line 229
Spelling: "Mis-assignment"
This is correct, the spelling was changed to the correct form throughout the article.
Reviewer 2 Report
Although poorly original, this study remains interesting, particularly because of the large cohort included for analysis. It is clear that the use of this type of tool can reduce the proportion of incorrectly sorted patients in the emergency department. However, in order to make the paper more original, it seems to me that including a more extensive analysis of the data would be necessary.
Did the researchers attempt to evaluate the economic gain to the hospital that this new procedure generated?
Did the researchers evaluate the impact of this new algorithm on the lenght of stay in the hospital or at the ermergency department ?
Researchers did they analyse the discrepant results between RT-PCR and RAD for Ct's values between 30 and 35 ?
Researchers precise that "the classification method of patients suspected of COVID-19, was based on a symptomatic case definition, as well as a history of exposure to a positive patient, or arrival from abroad or from hyperendemic sites within Israel." ; I suppose that the proportion of patients with a history of exposure to a positive patient was higher in the second period ? Should this bias not be discussed ?
L144: did the researchers refer to absolute or relative values ? Specify.
Author Response
We thank you for allowing us to revise the article, and we have carried out revisions following the reviewers’ comments. Please see our responses below, addressing each comment raised by each reviewer:
Reviewer 2 comments:
Although poorly original, this study remains interesting, particularly because of the large cohort included for analysis. It is clear that the use of this type of tool can reduce the proportion of incorrectly sorted patients in the emergency department. However, in order to make the paper more original, it seems to me that including a more extensive analysis of the data would be necessary.
We thank the reviewer for the general positive comment.
Did the researchers attempt to evaluate the economic gain to the hospital that this new procedure generated?
Regarding the economic gain to the hospital, we were unable to retrieve the desired data. However, we agree that this is an important aspect of real-world rapid tests usage, and we have now added a short paragraph to the discussion, addressing this subject, and referring to published literature on this topic (lines 267-273).
Did the researchers evaluate the impact of this new algorithm on the length of stay in the hospital or at the emergency department?
This is indeed a very important outcome, and it was added to the statistical analysis section (now lines 117-124), the results section (section 3.5), and the discussion section as well (lines 255-285). We also added Table 3 to further demonstrate the ED admission length changes.
Researchers did they analyze the discrepant results between RT-PCR and RAD for Ct's values between 30 and 35?
While Ct values >30 have been repeatedly shown to be less or non-infective, to address this comment, we have added a paragraph regarding mis-assigned patients with Ct values higher than 30 was added to section 3.3 - green ED risk reduction (now lines 174-182).
Researchers precise that "the classification method of patients suspected of COVID-19, was based on a symptomatic case definition, as well as a history of exposure to a positive patient, or arrival from abroad or from hyperendemic sites within Israel.”; I suppose that the proportion of patients with a history of exposure to a positive patient was higher in the second period? Should this bias not be discussed?
Yes, potentially, in the second period, more asymptomatic, exposed patients may have arrived at the red ER. However, we don’t see how this would bias the results. We have previously reported, that the sensitivity of Ag RDT was not different between asymptomatic and symptomatic patients with Ct value<30 [1]. Furthermore, as observed in Figure 1, the rates of infection between the two periods, were relatively similar.
L144: did the researchers refer to absolute or relative values? Specify.
At the beginning of this paragraph, we refer to absolute values – the average daily numbers of patients at each ED during both research periods. However, this trend (the non-significant change) also applies to the relative proportions of mis-assignation to the green ED. We specified this in this paragraph (now line 157), hoping that it is clearer now.
References:
[1] Regev-Yochay, G., Kriger, O., Mina, M.J., Beni, S., Rubin, C., Mechnik, B., Hason, S., Biber, E., Nadaf, B., Kreiss, Y. and Amit, S., 2021. Real world performance of SARS-CoV-2 antigen rapid diagnostic tests in various clinical settings. Infection Control & Hospital Epidemiology, pp.1-20.

Round 2
Reviewer 2 Report
I thank the authors for having taken into account my remarks and for having modified their manuscript. I think this new data reinforces the originality of the article.
The article is now acceptable, subject to some final (very) minor changes:
- Redrafting the abstract in light of the new results generated, particularly concerning the impact on the length of the hospital stay
- Line 100: Green and not Greed
Author Response
We thank the reviewer for the additional comments. The spelling error was corrected and the abstract was modified to describe the new results.
